# The Role of Functional Deficits, Depression, and Cognitive Symptoms in the Perceived Loneliness of Older Adults in Mexico City

**DOI:** 10.3390/ijerph21080977

**Published:** 2024-07-26

**Authors:** Ana Belén Ramírez López, Yaneth Rodríguez-Agudelo, Francisco Paz-Rodríguez, Silvia Aracely Tafoya, Benjamín Guerrero López, Claudia Diaz Olavarrieta

**Affiliations:** 1Mexican Institute of Social Security, Morelos Psychiatry Hospital, San Pedro el Chico, Gustavo A. Madero, Mexico City 07450, Mexico; anabelen.ramlo@comunidad.unam.mx; 2Research Division, National Institute of Neurology and Neurosurgery MVS, Mexico City 14269, Mexico; jrodriguez@innn.edu.mx (Y.R.-A.); fpaz@innn.edu.mx (F.P.-R.); 3Department of Psychiatry and Mental Health, Faculty of Medicine, National Autonomous University of Mexico, Mexico City 04510, Mexico; stafoya@unam.mx (S.A.T.); dr.bguerrero@unam.mx (B.G.L.)

**Keywords:** loneliness, functionality, mild neurocognitive disorder

## Abstract

The world is aging and experiencing loneliness. Functional impairment in instrumental activities of daily living (IADL) in older people (OP) with mild neurocognitive disorder (MNCD) predicts loneliness. After the pandemic, there was an increase in perceived loneliness. We explored the association between loneliness, depression, deficits in IADL, and cognitive symptoms among OP. From February to December 2023, using a cross-sectional design, we interviewed probable cases with mild cognitive impairment and caregivers in two public facilities. We administered the UCLA Loneliness Scale v3, Lawton IADL Scale, Mini-Mental State Examination (MMSE), and Yesavage’s Geriatric Depression Scale. Samples were matched: 85 per group, 82.4% were women, married (52.95%), and mean age of 69.17 (±6.93) years. In our study, 30% displayed moderate to high levels of perceived loneliness. Multivariate analysis showed loneliness was associated with depression, low levels of IADL, and older age, but not with cognitive symptoms, which explained 22% of the total variance (F 165) = 16.99, (*p* < 0.001). Targeting symptoms and behaviors that could be modified (i.e., depression and functionality) can improve feelings of perceived loneliness and have an impact on morbidity and mortality with which it is associated.

## 1. Introduction

The aging of the world population constitutes a significant public health concern, as the percentage of people over 60 years old will double between 2015 and 2050 [1]. Mexico’s official statistics estimate that between 2020 to 2025, this age group (60+ years) will grow by 3.5% and, according to data from the United Nation’s Economic Commission for Latin America and the Caribbean (ECLAC), the regional growth between 2010 and 2060 will reach 10% [2,3]. 

According to the World Health Organization (WHO), the experience of loneliness among older people (OP) is a widespread condition; between 20% and 34% of OP in China, Europe, Latin America, and the United States feel lonely [4]. Research comparing global prevalences display significant variability, with prevalences ranging from 2% to 50% [5]. Prior to the start of the COVID-19 pandemic, the National Study of Health and Aging in Mexico reported that most of the elderly population had social and family networks, with only 9.5% of those aged 60–79 years and 15.1% aged ≥ 80 years living alone. In this same age group, 91.3% and 90.8% were in touch with their children; in addition, 58.9% and 56.2% reported receiving support from neighbors. However, the percentage mentioning feelings of loneliness was high for both groups, at 41.8% and 43.6%, respectively [6]. After the start of the pandemic, Mexico’s National Health and Nutrition Survey noted that 12.3% of people aged ≥ 65 were living alone (vs 11.4% before the pandemic), with a high prevalence of social isolation (participants who were not in contact with others via mobile phones or electronic devices, 45.3%) and loneliness (almost 50%) reported among people living alone; however, the survey did not explore the perception of loneliness among those not living alone [7]. Age and social isolation, the new “geriatric giants”, are associated with the perception of loneliness [8,9]. 

Loneliness, defined as the quality of a person’s relationships with others, is a “distressing feeling that accompanies the perception that one’s social needs are not being met by the quantity or especially the quality of one’s social relationships” [10]. Loneliness is a complex and multifactorial experience, currently considered an index of quality of life and health, which can increase the risk of mortality from any cause by up to 45% among people aged ≥ 60 years, and it can also significantly increase the risk of developing diseases including cardiovascular disease, cognitive impairment, functional impairment, and psychiatric disorders [11,12,13,14,15].

The SARS-CoV-2 pandemic abruptly imposed restrictive and isolation measures on the world’s social interaction [16,17]. After countries adopted these measures, an increase in loneliness was observed among up to one third (33.7%) of the elderly for whom this was an entirely novel and disquieting experience. Furthermore, the probability of experiencing loneliness showed a twofold increase among those who lived alone compared to those who lived as a couple [18]. 

Limited functionality in instrumental activities of daily living (IADL) has been considered a predictor of loneliness and social isolation; as the limitations increase, there is a greater experience of loneliness and greater distance from others. Research among OP, including those more at risk of developing dementia and those with limitations to carry out IADL, have shown they have less social support, a decreased social network size, and lower participation in social activities than those without such deficits or limitations [19,20]. Functionality is a complex condition influenced by cognitive, psychological, physical, and environmental factors [21]. Previously it was believed that people with mild neurocognitive disorder (MNCD) would not display significant functional deficits; however, deficits in at least one instrumental life activity daily have shown prevalences of up to 80% among subjects with MNCD [22,23]. Depending on which domains are involved, different cognitive functions will also be compromised; in addition, limitations in activities of daily living have also been shown to be a statistically significant predictor of incident dementia [24]. Regarding functional limitations in IADL in Mexican OP, in 2015 14.6% of the population > 60 years displayed some type of deficit or limitation, with differences according to sex: it was found in 18.5% of women and 10% of men [25]. A study in the U.S. showed IADL impairments in 31.3% of people ≥ 65 years, women with impaired IADL had a prevalence of 29.8%, and men had a prevalence of impairment of 20.2%. The degree of impairment found in Mexican women was similar to that of other countries [26,27].

Studying the association between loneliness, depression, functionality, and outcomes in social resources among OP in Mexico, controlling for participants with probable mild neurocognitive disorder and healthy senior citizens, is of paramount importance. The elderly, especially those with neurocognitive symptoms, are highly susceptible and more at risk of experiencing loneliness [28]. Exploring its impact on this age group using standardized measuring instruments that are easy to administer could allow us to characterize this phenomenon more thoroughly. It will permit a more multifaceted understanding of the ways in which these variables interact, and we will be able to draft a comprehensive assessment protocol that will lead to multidisciplinary interventions with the aim of strengthening the elderly’s social support networks and improving the physical and mental health of patients seeking care at a public tertiary reference center for neurological diseases (National Institute of Neurology and Neurosurgery, NINN) and those treated in the psychogeriatric clinic of third-level care at a public psychiatric hospital in Mexico City.

The present study is based on two hypotheses: (1) we will observe a high prevalence of loneliness among older people; and (2) the extent of loneliness will be more significant in participants displaying a greater number of depressive symptoms, a higher prevalence of cognitive impairment, lower levels of functioning in daily activities, advanced age, and a lack of a partner (i.e., living alone).

## 2. Materials and Methods

### 2.1. Participants and Procedure

Ours was a cross-sectional study carried out in two public third-level-of-care facilities. We invited older people seen in the outpatient clinic of a psychogeriatric clinic for suspected cognitive impairment, patients with probable neurological deficits, and elderly caregivers of neurological patients treated in a neuropsychology clinical laboratory during February–December 2023. We did not establish other inclusion or exclusion criteria; all patients and caregivers were aged ≥ 60 years. Once they agreed to participate, they signed informed consent forms. A member of the study team then confirmed they were able to answer the study surveys, and they were recruited as participants.

The study was approved by the bioethics and research committees of both study sites. Once we obtained the informed consent of each participant, we collected sociodemographic data, and the study instruments were administered during the consultation hour assigned to each participant. 

For both groups, the UCLA Loneliness Scale and the PHQ-9 questionnaire were self-administered, while the Yesavage Geriatric Depression Scale, the Lawton Index, and the Mini-Mental State Examination (MMSE) were administered by a trained researcher. A total of 86 patients were invited to participate from the psychogeriatric clinic, one participant was excluded due to missing data, and the final sample size included 85 patients. 

Because our samples were recruited from two study sites and showed disparate sociodemographic characteristics, we needed to ensure both samples were appropriately matched by sex, age, and marital status. This was done to prevent any potential confounding effects associated with these variables, allowing for a more nuanced examination of our primary independent variables, including probable depression, MMSE scores, and functional capacity. We thus obtained a total of 85 participants in each group (those with probable neurological deficits and healthy subjects). Lastly, the groups were treated as a single entity to analyze the impact of the primary independent variables.

### 2.2. Measurements

#### 2.2.1. Loneliness

The University of California Los Angeles (UCLA) Loneliness Scale version 3 [29], includes 20 items with four response options that are rated on a Likert-type scale ranging from 1 = “Never” to 4 = “Always”. The UCLA Loneliness Scale includes eleven items that indicate the presence of loneliness and nine referring to its absence (they are calculated in reverse); the total score is made by adding the scores of each item. This scale is widely used in research to evaluate the degree of loneliness, and it does not establish a clinical cut-off point that indicates significant levels of loneliness. However, the following cut-off points have been proposed: scores of 20 to 34 indicate a low degree of loneliness; scores of 35 to 49 indicate a moderate degree of loneliness; scores of 50 to 64 indicate a moderately high degree of loneliness; and scores of 65 to 80 points indicate a high degree of loneliness [30,31]. For this study, we administered the Loneliness Scale v3 in Spanish for Mexico. The instrument has shown high internal consistency (alpha coefficient = 0.96) and a test–retest correlation over a two-month period of 0.73 [32].

#### 2.2.2. Functionality

The Lawton Instrumental Activities of Daily Living Scale, developed at the Philadelphia Geriatric Center, was used to assess physical autonomy and IADL in inpatient or outpatient settings [33]. The index is used to identify the degree of independence of OP in carrying out IADL, and it assesses eight activities: using the telephone, cooking, washing clothes, doing housework, using transportation, managing finances, making purchases, and managing medications. Each item is assigned a numerical value, 1 = “independent” or 0 = “dependent”, and the final score is obtained from the sum of the values of all the responses, ranging between 0 and 8, where 0 indicates maximum dependence and 8 total independence. In the Spanish population, it showed a Cronbach’s alpha coefficient of 0.94 and factor loadings of 0.67 to 0.90, confirming the homogeneity of the construct [34].

#### 2.2.3. Cognitive Impairment

To detect the presence of probable cognitive impairment, we used the Mini-Mental State Examination (MMSE), a screening test for neurocognitive disorders [35]. The test includes 19 items that assesses six cognitive domains: spatial and temporal orientation, fixation memory, evocation memory, attention, calculation, and language. To obtain a rating, the number of correct answers in the tests is counted and a higher score indicates an unimpaired cognitive state. The MMSE is also used to detect cognitive decline, determine the severity of cognitive decline if this is present, and monitor a person’s cognitive changes. We used the version adapted and validated into Spanish [36], which considers sociodemographic variables such as age and level of educational attainment. A sensitivity of 97% and specificity of 88%, with an area under the curve of 0.85, has been reported to identify cognitive impairment. In our study, the MMSE cut-off score to determine cognitive decline was 24.

#### 2.2.4. Confounding Variables

Considering sociodemographic characteristics, we enquired about sex, age, level of education, and marital status.

#### 2.2.5. Probable or Confirmed Current Major Depressive Episodes

In the psychogeriatric clinic participants, the diagnosis of current depression was based on the psychiatric interview of the Diagnostic and Statistical Manual of Mental Disorders, Fifth Edition (DSM 5-TR) [37] and Yesavage’s 15-item Geriatric Depression Scale (GDS-15) criteria [38]. Among the participants who were caretakers of neurological patients and were included in the study, the probable presence of current depression was evaluated using the Patient Health Questionnaire (PHQ-9), which is a self-administered questionnaire with nine items that provides a quantitative assessment of the severity of depressive symptoms at the time it is administered. Its score is obtained by adding the responses of each item [39]. Traditionally, a score ≥ 10 is considered to indicate a probable episode of major depression [40]. Although we used two different measuring instruments to evaluate the presence of depressive symptomatology, research has confirmed both are comparable. The GDS-15 and PHQ-9 have comparable diagnostic accuracy in classifying older adults with depressive episodes [41,42].

### 2.3. Data Analysis

The data was analyzed using the SPSS statistical program, v.23.0 (IBM Corp. Armonk, NY, USA). An α < 0.05 was considered statistically significant. The Kolmogorov–Smirnov test was applied to verify the normality of the data. In addition, the omega coefficient of the instruments was calculated with the JASP 14.0.01 statistical package [43]. Descriptive statistics were used to characterize the sample, and bivariate analysis was performed to compare the groups: chi-squared or Fisher test for qualitative variables; *t* test or Mann–Whitney U test for quantitative variables. To assess the association between the study variables, we used Spearman correlation analysis. Lastly, we used multiple linear regression analysis to determine the predictors of loneliness, including depression and MMSE and IADL scores as independent variables, and the demographic characteristics of the study participants were also included: sex, marital status, whether they had a partner or not, study group (those with probable neurological deficits and healthy subjects), and level of schooling. We performed the regression model only with predictors that correlate with the dependent variable. We assumed no collinearity when the values of the variance inflation factors were less than 4.0 and the tolerance factors were greater than 0.2, and a Durbin–Watson residual close to 2 was observed to consider that no autocorrelation occurred (Hair et al., 2006] [44]. Outliers were determined with a z score greater than 3 or less than −3 (see Figure 1). The alpha–omega value for the loneliness scale was 0.924 and the alpha–omega value for the IADL scale was 0.940.

## 3. Results

### 3.1. Characteristics of Study Participants

The samples were matched according to some of their demographic characteristics (sex, age, and marital status), resulting in a total of 85 participants in each group (Table 1). In general, there was a higher proportion of women (82.4%), a higher proportion of married participants (52.95%), and a mean age of 69.17 years. The following differences were observed between the groups, the sample treated in the psychiatric hospital and the one seen at the clinical neuropsychology laboratory (NINN), respectively: the frequency of cases with depression was 35.3% and 10.6%; the level of schooling had an average of 6.75 and 8.74 years; the degree of cognitive impairment showed a score of 23.87 and 25.74 in the MMSE; the functionality in IADL was 5.88 and 7.51 points; and finally, for the degree of loneliness the scores were 50.27 and 34.55 points. We noted that 68% of all participants displayed some degree of loneliness, but 30% of the total sample had a moderately high to high degree of perceived loneliness.

### 3.2. Spearman Correlations

When analyzing correlations (Table 2), we noted a statistically significant, low, and negative correlation between loneliness and level of schooling, sex (being female), group (probable neurological deficits), and MMSE and IADL scores. Furthermore, as expected we also observed a low significant and directly proportional correlation between perceived loneliness and depression. We also found a moderate negative correlation between age and IADL. Depression showed a low negative correlation with MMSE and IADL scores.

### 3.3. Multiple Regression Model for Perceived Loneliness in Older Adults and Association between Model Variables

Lastly, Table 3 shows the multiple linear regression analysis where the predictor variable of perceived loneliness was associated with the group they were recruited into (probable neurological deficits) and symptoms of depression. The adjusted R2 coefficient indicates our model explained 36% of the total loneliness variance. The F value in the analysis of variance (F 166) = 48.94, (*p* < 0.001) shows the existence of a significant linear association between group (probable neurological deficits) and depression as the main variables of the model.

## 4. Discussion

Ours was a cross-sectional study carried out in two public third-level-of-care facilities among 85 older adults who were seen in the outpatient clinics of a psychogeriatric clinic for suspected cognitive impairment and 85 participants who were caregivers of neurologic patients (older adults), all seen during February–December 2023. We explored the association between perceived loneliness, depression, deficits in daily functioning, and cognitive symptoms among OP, hypothesizing that all of these would be predictors of a greater perception of loneliness. Our results indicate a high prevalence of loneliness among participants, as well as a greater perception of it among those with depressive symptoms, decreased degree of daily functioning and being older; however, the perception of loneliness was not associated with cognitive symptoms. To our knowledge, this is the first attempt at studying the effects of depression and level of functionality on older adults’ perception of loneliness post COVID-19 among this age group in Mexico City.

Regarding our first hypothesis, we found that a third of the study participants surveyed displayed levels of perceived loneliness in the moderate to high range. Previous studies worldwide, reported through a systematic review, showed higher prevalences of loneliness in OP compared to younger ones. The prevalence of loneliness in groups of OP in European countries fluctuated between 4.2% and 24.2%, with lower levels of loneliness found among those with higher socioeconomic status, improved health status, greater welfare generosity, and high social participation [45]; challenging conditions that are often not met in low- or middle-income countries such as Mexico. The level of perceived loneliness observed in our study was like that reported during the COVID-19 pandemic [18]; however, it was also similar to the level reported in people over 50 years of age in the US [13]. 

The second hypothesis proposed several factors that may be associated with the experience of loneliness. In our study, we observed higher levels of loneliness associated with greater difficulties in functioning and greater affective (depressive) symptoms, and a slight association with the presence of cognitive symptoms, in accordance with criteria suggested by Surkalim et al., 2022 [45]. We also believe that the sudden confinement imposed by the pandemic may have increased the participants’ perception of loneliness, since health and isolation conditions may not have returned to their pre-pandemic levels, an area that merits further research.

It is worth noting that the association found between deficits in daily functioning and loneliness has been previously reported. One study indicates that perceived loneliness can be associated with a decrease in physical activity mediated by marital status, since widowed people with higher rates of loneliness seem to exhibit decreased functioning more than those who are married or separated [46]. While other studies point out that these findings depend on sex, it is an association observed only in women, but not in men [11,13].

These studies indicate a directionality that ranges from loneliness to functioning, but a systematic review pointed out that this relationship can also be bidirectional [12]. Although our study, due to its cross-sectional nature, does not allow us to establish a causal relationship, we think there may be factors that influence each other. Regarding the association found between depression and perception of loneliness, this finding has also been reported consistently. Again, most studies point to loneliness as a precursor of depression [47,48], but there is also evidence that shows that feelings of loneliness can contribute to the presence of depression [49]. Likewise, it has been suggested that depression, when associated with cognitive symptoms, increases loneliness in OP [50]. Therefore, we can put forth a bidirectional association between these variables, which can become a vicious circle that has an increasingly negative effect and decreases the patient’s sense of well-being. We also noted that the MMSE scores were correlated with loneliness, but the multiple regression model did not find a significant association. This finding is contrary to what has been reported in the literature [51]; however, we think the symptoms of depression could have explained these two variables, or the patients’ symptoms could have been mild, and thus we were not able to observe its effect. 

Among the study limitations, we find that our cross-sectional study design did not allow us to infer causality among study variables, and even though we were able to pair our samples (by sex, age and marital status) to make them homogeneous, we had to carry out convenience sampling. A further limitation was the use of different instruments to assess probable depression, because participants came from two disparate public institutions. While both scales are internationally recognized and have reliability, validity, and well-established cutoff points for depression in our population, their scores are not directly comparable. Consequently, they can only be considered as indicators of the presence of significant depression symptoms. A study strength was that we were able to sample patients and healthy controls from two of the most underprivileged districts of a mega-city (Mexico City), who had some of the highest COVID-19 morbidity and mortality rates and thus could be subject to experiencing higher rates of perceived loneliness due to the limited home, work, and family infrastructure imposed during confinement [18,52]. 

## 5. Conclusions

Loneliness has been shown to increase the morbidity and mortality of older adults [9]. Data mostly from industrialized countries is consistent with our study findings, so there is a pressing need to implement strategies for first-level-of-care physicians to identify lonely patients. While our study was fielded in tertiary-care public facilities, we could extrapolate and perhaps increase our prevalences when we consider the patient population normally seen in primary care. It seems that Mexico’s epidemiological transition now includes frail, elderly people living in underprivileged conditions, such as our study participants, who reported a heightened perception of loneliness and mental health indicators that are consistent with the need to implement strategies to help reduce this new giant, even in countries such as ours with a myriad of competing health inequities.

## Figures and Tables

**Figure 1 ijerph-21-00977-f001:**
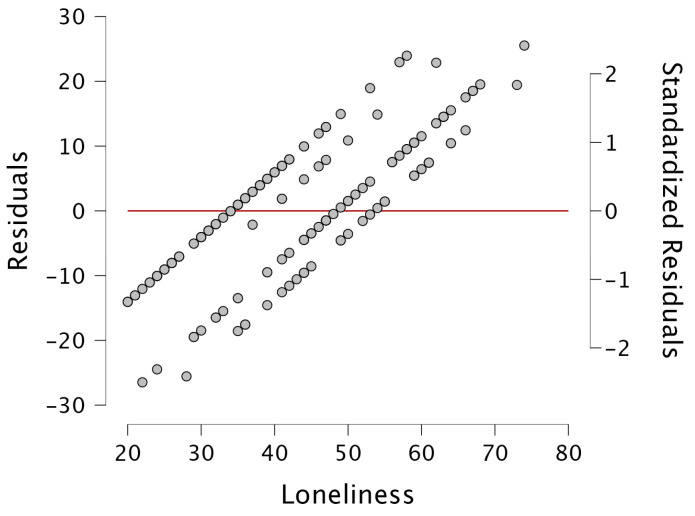
Loneliness vs. standardized residuals. Information on possible outliers.

**Table 1 ijerph-21-00977-t001:** Characteristics of study participants.

	Group		Total	*p*
	Participants with Probable Neurological Deficits	Healthy Participants (Caregivers)		
	n (%)	n (%)	n (%)	
Sex				
Women	70 (82.4)	70 (82.4)	140 (82.3)	
Men	15 (17.6)	15 (17.6)	30 (17.7)	1.000
Marital status				
Married/common law	40 (47.1)	50 (58.8)	90 (52.9)	
Widowed	29 (34.1)	24 (28.2)	53 (31.2)	
Divorced	8 (9.4)	2 (2.4)	10 (5.9)	
Single	8 (9.4)	9 (10.6)	17 (10.0)	0.155
Depression				
Yes	30 (35.3)	9 (10.6)	39 (22.9)	
No	55 (64.7)	76 (89.4)	131 (77.1)	<0.001
Degree of perceived loneliness				
Low	9 (10.6)	45 (52.9)	54 (31.8)	
Moderate	31 (36.5)	34 (40.0)	65 (38.2)	
Moderately high	38 (44.7)	6 (7.1)	44 (25.9)	
High	7 (8.2)	0 (0)	7 (4.1)	<0.001
	Mean ± SD	Mean ± SD	Mean ± SD	
Age (years)	71.47 ± 7.53	70.27 ± 6.92	69.17 ± 6.93	0.340
Level of schooling (years)	6.75 ± 3.59	8.74 ± 4.84	9.02 ± 5.07	0.009
MMSE (score)	23.87 ± 2.72	25.74 ± 2.73	25.28 ± 2.83	<0.001
IADL (score)	5.88 ± 2.15	7.51 ± 1.11	7.06 ± 1.67	<0.001
Loneliness (score)	50.27 ± 11.86	34.55 ± 9.59	38.82 ± 12.72	<0.001

Note: Comparisons were made using chi-squared/Fisher test and Mann–Whitney U test. IADL, Instrumental activities of daily living. MMSE, Mini-Mental State Exam.

**Table 2 ijerph-21-00977-t002:** Spearman correlations.

Variable	1	2	3	4	5	6	7	8
1. Perceived loneliness	—							
2. Age	0.109	—						
3. Level of schooling	−0.273 **	−0.279 **	—					
4. Depression	0.311 **	0.076	−0.122	—				
5. Sex	−0.292 **	−0.080	0.209 **	−0.115	—			
6. Group	0.594 **	0.073	−0.201 *	0.294 **	0.000			
7. MMSE	−0.286 **	−0.207 **	0.335 **	−0.194 **	0.148 *	0.355 **		
8. IADL	−0.396 **	−0.421 **	0.298 **	−0.185 **	0.125 *	0.460 **	0.382 **	—

Notes: IADL, Instrumental activities of daily living. MMSE, Mini-Mental State Exam. ** The correlation was significant at alpha level of 0.01; * The correlation was significant at alpha level of 0.05.

**Table 3 ijerph-21-00977-t003:** Multiple regression model for loneliness in older adults (n = 166).

Variable	Unstandardized	Standard Error	B	*T*	*p*	95% CI
(Intercept)	62.90	2.86		22.02	<0.001	60.76, 71.13
Group (probable neurological deficits)	14.43	1.71	0.54	8.43	<0.001	11.05, 17.81
Depression	5.08	2.03	0.16	2.502	0.013	1.07, 9.10
F	48.94					
Adjusted R^2^	0.36					
Durbin–Watson	1.72					

Note. The following covariates were considered but not included: level of schooling, sex, does not live with a partner, and MMSE scores.

## Data Availability

The original contributions presented in this study are included in the article. Further inquiries can be directed to the corresponding author.

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
