# Peer review of "The Role of Functional Deficits, Depression, and Cognitive Symptoms in the Perceived Loneliness of Older Adults in Mexico City"

_ijerph, 2024, doi:10.3390/ijerph21080977_

Round 1

Reviewer 1 Report

Comments and Suggestions for Authors

Good morning,

thank you for providing this article. I liked the introduction very much, since it was clear and showed the importance of the study. Only validated instruments were used to guarantee comparability to other studies. Some important clarifications should be made:

1. Please provide more details about data collection. Did patients fill in the questionnaire themselves or were they supported by any specialists? Was the data collection method the same in both groups? Did you analyse all questionnaires or was there a certain drop out (i.e. missing data, refusal to participate...)? Was the recruitment method the same in both facilities? Despite the 2 different scales for depression, were there any other differences between the questionnaires of the 2 groups?

2. You used two different questionnaires for depression. Can you provide any reference to guarantee comparability of the two instruments? Since the former questionnaire is declared as an instrument to measure depression, the PHQ-9 only provides non clinical information about depressive symptoms. Even if results in table 1 indicate a clear difference between the 2 groups, I wonder how you can provide information about a significant difference between the percentage of depression. It would be better only to present the percentages without a p-value.

3. Please provide a sample size estimation for the regression model

4. Even if age was not correlated to loneliness, you used it in the regression model and it resulted as a significant predictor. This seems to be due to correlation with other predictors. I would propose to recalculate the regression model only with predictors that correlate with the dependent variable. Further, regression diagnostics should provide information about possible outliers.

5. I would appreciate if you list your hypotheses at the end of the introduction as a list 1., 2... and structure your discussion in the same way in order to discuss each hypothesis clearly. At the moment the discussion is difficult to comprehend.

6. On line 305 of the discussion you write you were able to pair your samples. What does this mean? Until this point I have thought that you had 2 independent groups.

Comments on the Quality of English Language

Please let proofread the text by a native speaker or any program providing English correction tools.

Reviewer 2 Report

Comments and Suggestions for Authors

The article on the mental health of the elderly is quite significant. However, there are a few points in the current article that are puzzling:

 1. The title uses the term "aftermath of the COVID-19 pandemic," but from the data collection in the article, it mainly involves standardized tools. Readers find it difficult to see from the article what the respondents experienced during the pandemic and what impact the transition from the pandemic to the post-pandemic period had on them.

 2. The article uses the terms "case" and "control" in the data description, but this paper is not an intervention study, but rather a cross-sectional study on influencing factors. All data used in the subsequent models are used together. It is suggested that the author provides further clarification on this.

Round 2

Reviewer 1 Report

Comments and Suggestions for Authors

ok for me

Reviewer 2 Report

Comments and Suggestions for Authors

The author has addressed my queries and made corresponding revisions to the article. I have no further questions and recommend that the paper be published.